**Data Availability Statement:** All relevant data are within the manuscript.

**Funding:** This work was supported by Animal Feed Inter Trade Co., Ltd, Thailand Science Research

# Bioconversion of agro-industrial residues as a protein source supplementation for multiparous Holstein Thai crossbreed cows

Chaichana Suriyapha[1], Chanadol Supapong[2], Sarong So[1,3], Metha Wanapat[1], Anusorn Cherdthong[1]*

1 Tropical Feed Resources Research and Development Center (TROFREC), Department of Animal Science, Faculty of Agriculture, Khon Kaen University, Khon Kaen, Thailand, 2 Department of Animal Science, Faculty of Agriculture, Rajamangala University of Technology Srivijaya, Nakhon Si Thammarat Campus, Nakhon Si Thammarat, Thailand, 3 Department of Animal Science, Faculty of Agriculture and Food Processing, National University of Battambang, Battambang, Cambodia

* anusornc@kku.ac.th

## Abstract

The purpose of this field study was to compare the effects of top-dressing tropical lactating cows with soybean meal (SBM) or citric waste fermented yeast waste (CWYW) on intake, digestibility, ruminal fermentation, blood metabolites, purine derivatives, milk production, and economic return. Sixteen mid-lactation Thai crossbreeds, Holstein Friesian (16.7 ± 0.30 kg/day milk yield and 490 ± 40.0 kg of initial body weight) were randomly allocated to two treatments in a completed randomized design: SBM as control (n = 8) or CWYW (n = 8). The feeding trial lasted for 60 days plus 21 days for treatment adaptation. The results showed that total dry matter intake, nutrient intake, and digestibility did not (p>0.05) differ between SBM and CWYW top-dressing. Ruminal pH and the protozoal population did not (p>0.05) differ between SBM and CWYW top-dressing. After 4 hours of feeding, CWYW top-dressing showed greater ammonia nitrogen, plasma urea nitrogen, and bacterial population compared with the top-dressing of SBM. Volatile fatty acids and purine derivatives were not different (p>0.05) between SBM and CWYW top-dressing. For milk urea nitrogen, there was a greater (p<0.05) and somatic cell count was lower (p<0.05) for cows fed the CWYW top-dress compared to cows fed the SBM top-dress. The cost of the top-dress and total feed cost were less (p<0.05) for CWYW compared to SBM top-dressing, at 0.59 *vs* 1.16 US dollars/cow/day and 4.14 *vs* 4.75 US dollars/cow/day, respectively. In conclusion, CWYW could be used as an alternative protein source to SBM without having a negative impact on tropical lactating cows.

## Introduction

Over 80% of milk in developing countries comes from small-scale farming operations with low inputs [1]. The availability and quality of feed are the two most important factors influencing milk production [2]. Rural smallholder dairy cow farmers in Thailand feed concentrate to their cows as an energy and protein source, while rice straw (RS) is given to them as a roughage

and Innovation (https://www.tsri.or.th) through the Research and Researcher for Industry (RRi) (http://rri.trf.or.th/bs_detail.asp) program (contract grant PHD62I0021), Research Program on the Research and Development of Winged Bean Root Utilization as Ruminant Feed, Increase Production Efficiency and Meat Quality of Native Beef and Buffalo Research Group and Research and Graduate Studies, Khon Kaen University (KKU), and Office of National Higher Education Science Research and Innovation Policy Council (https://www.nxpo.or.th/) through the Program Management Unit for Competitiveness (PMUC) (https://www.nxpo.or.th/) (contract grant C10F640078). Funders did not play any role in the study design, data collection and analysis, decision to publish, or preparation of the manuscript.

**Competing interests:** The authors have declared that no competing interests exist.

source on an ad libitum basis. RS is a low-nutritive, high-indigestible fiber roughage with long particles that can quickly fill the rumen, resulting in decreased feed intake and insufficient nutrient supply for the host [3]. Farmers always top-up soybean meal (SBM) on concentrate at approximately 0.4 percent body weight to maximize or maintain milk production. SBM is the most commonly used protein source in dairy cow concentrate mixes [4, 5] and is high in rumen-degradable protein [6]. According to Ghelichkhan et al. [7], a top-dress of 2 kg of solvent SBM can increase milk output by 7.2 kg/day in Holstein Friesian cows. However, because SBM is in high demand, the price has risen [8, 9]. Furthermore, increased soy cultivation has negative environmental and socioeconomic consequences [10]. Currently, there is a global interest in using agro-industrial by-products and residues as animal feed, utilizing biological processes as ecologically acceptable and potentially economically feasible options for decreasing feed prices and alleviating the problem of environmental contamination [11–14].

Citric acid waste fermented with yeast waste (CWYW) is a feed product made from citric acid waste from the citric industry fermented with yeast waste from bio-ethanol processing as an alternative protein source for ruminants [14]. With a crude protein content of 53.5% and a true protein content of 33.5%, CWYW is a promising protein source comparable to SBM [14]. When comparing economic values, CWYW was cheaper than SBM by 50% per kilogram (0.31 USD *vs* 0.61 USD). Furthermore, Suriyapha et al. [14] discovered that when CWYW replaced SBM at 75% of SBM in a concentrate diet. Gas kinetics, ruminal fermentation, and *in vitro* degradability were unaffected. The comparative effect of top-dressing SBM and CWYW on dairy cow feed intake, digestibility, ruminal fermentation kinetics, milk yield, and milk composition has never been studied.

As a result, the purpose of this field study was to compare the effects of top-dressing SBM or CWYW on the intake, digestibility, ruminal fermentation, blood metabolites, purine derivatives, milk production, and economic return of tropical lactating cows.

## Materials and methods

All of the experimental animals and methodology involved in this research were approved by the Animal Ethics Committee under the Institutional Guidelines of Khon Kaen University, National Research Council of Thailand (record no. IACUC-KKU-27/64).

### Preparation of citric waste fermented yeast waste (CWYW)

A by-product of ethanol manufacturing called yeast waste, which includes inoculants of the yeast (*Saccharomyces cerevisiae*), was obtained from Green Innovation Public Company Limited (KGI) a subsidiary of Khon Kaen Sugarcane Industry Company Limited in Nam Phong district, Khon Kaen province, Thailand. Citric acid waste, a by-product of the citric acid industry, was obtained from Sam Mor Farm Limited Partnership in Mueang district, Udon Thani province. The commercial feed pellets, commercial-grade urea, and molasses were purchased from the local feed mill in the Kranuan district, Khon Kaen province, Thailand.

CWYW was prepared according to Suriyapha et al. [14]. In brief, 100 mL of yeast waste was added into a flask and set as solution A. Then, 20 g of molasses and 50 g of urea were dissolved in 100 mL of distilled water to form solution B. After that, solutions A and B were mixed at a ratio of 1:1 (v/v). The solution's pH was adjusted to 3.5–5 using formic acid (L.C. Industrial Co., Ltd, Nakhon Pathom, Thailand) and incubated to stimulate yeast population growth with continuous air-flush for 16 h using an air pump (HAILEA ACO-318 oxygen pump, Sagar aquarium, Gujarat, India) at room temperature. After 16 h of incubation, the media solution of yeast waste was mixed with citric acid waste at a ratio of 1:1 (v/w). Then the mixture was anaerobically fermented in 200-liter plastic containers for 14-days, followed by 48 h of sun

drying to obtain a moisture level of less than 10%. After drying, the CWYW was kept in plastic bags and fed to the cows throughout the experimental period.

## Experimental station

The experiment was conducted at Sukkhee farm, a dairy farm in Kranuan District, Khon Kaen Province, Thailand. The experiment lasted from September 2020 to November 2020 during the rainy and winter season, with a temperature of approximately 22–33 ˚C. Before starting the experiment, cows were of a similar age and lactation yield period. All of the cows were weighed, recorded as their initial body weight, and dewormed (Ivomec F®, Kos Introtech Co., Ltd., Bangkok, Thailand) and injected with vitamin $AD_3E$ (Phenix, Anitech Total Solution Co., Ltd., Bangkok, Thailand).

## Experimental animals, design, and dietary treatments

Sixteen multiparous Holstein Thai crossbreed cows during mid-lactation with 16.7 ± 0.30 kg day-in-milk and 490 ± 40.0 kg BW were used. Cows were randomly divided into two groups and assigned to receive two supplementary treatments: the control-SBM group (n = 8) and CWYW (n = 8). Cows were individually housed in sixteen 5×5 $m^2$ stalls with cement water tanks. The SBM top-dress was used as the control treatment based on the routine practices used at the Sukkhee Farm. The SBM and CWYW were top-dressed onto a basal diet at a rate of 0.4 kg/100 kg of cow BW per day, followed by *ad libitum* rice straw feeding. The basal concentrate diet contained 26% cassava chip, 25.5% rice bran, 14% cassava leaf hay, 17% commercial pellet, 17% commercial protein mixed powder, and 0.5% mineral premix. Cows were fed the basal diets at a 1:1 ratio (1 kg concentrate per 1 kg milk) in two equal meals at 5:30 am and 3:30 pm. The commercial pellet was purchased from Chok Prasert Animal Feed Co., Ltd., and contained 25% crude protein (CP) based on a dry matter basis, while the commercial protein mixed powder was purchased from CPF Feed Solution Co., Ltd. and contained 46% CP on a dry matter basis. The urea concentration was less than 2% in the pellet and protein mixed power. The cows were adapted to stalls and diets for 21 days, and the supplementation treatments lasted for 60 days after adaptation. The chemical composition of the basal diet, SBM, CWYW, and rice straw is provided in Table 1. Feed intake, feed refusal, and milk yield were recorded daily, and the last 7 days of the supplementation period were used for sample collection.

**Table 1. Chemical compositions of experimental dietary.**

| Item | Basal concentrate diet | SBM[1] | CWYW[2] | Rice straw |
|---|---|---|---|---|
| Dry matter, g/kg | 875 | 890 | 864 | 909 |
| Organic matter, g/kg DM | 916 | 933 | 925 | 907 |
| Ash, g/kg DM | 84 | 67 | 75 | 93 |
| Crude protein, g/kg DM | 163 | 458 | 528 | 27 |
| Neutral detergent fiber, g/kg DM | 198 | 173 | 392 | 784 |
| Acid detergent fiber, g/kg DM | 143 | 133 | 281 | 552 |
| Feed cost (US dollar[3] /kg) | 0.38 | 0.61 | 0.31 | 0.07 |

[1]*SBM*, soybean meal;

[2]*CWYW*, citric waste fermented yeast waste;

[3] *US dollar* = 31.16 Thai baht [78].

## Sample collection and chemical analysis

During the 7 day sample collection period, the basal diet, rice straw, SBM, CWYW, and refusal samples were collected daily, composited by cow, and stored at -20 ˚C until analysis. Feces were collected manually from the rectum in the morning and in the afternoon before feeding for 7 days, weighed, 300 g composited by cow, and stored at -20 ˚C. Before analysis, diet samples were thawed, dried at 60 ˚C for 72 hours, and ground through a 1 mm screen using a Cyclotech Mill (Tecator, Hoganas, Sweden). Chemical composition was analyzed according to the standard method of AOAC [15] including dry matter (DM, no 967.03), ash (no 492.05) and crude protein (CP, no 984.13) content. The neutral detergent fiber (NDF) and acid detergent fiber (ADF) content of samples were analyzed according to the procedure of Van Soest et al. [16]. Urine was collected manually by stimulating the vulva twice a day in the morning and in the afternoon before feeding for 7 days. The urine sample from each day was composited by cow, kept in a plastic bottle containing $H_2SO_4$ at a 1:9 (v/v) ratio to prevent nitrogen loss, and stored at -20 ˚C until analysis. The urinary sample was thawed and analyzed for total purine derivatives, allantoin, and creatinine concentrations using high-performance liquid chromatography, and the effluent was monitored at 205 nm (Spherisorb; Agilent 1100 Series HPLC System Agilent Technologies, USA) with two 4.6 mm × 250 mm C18 reverse phase columns (Waters, Milford, MA, USA). Daily urine volume was estimated as BW × 29/urinary creatinine concentration [17]. Urinary purine excretion, purine absorption, and microbial nitrogen supply were estimated using the equations of Chen and Gomes [18] and Chen et al. [19]: $Y = 0.85X + (0.385BW^{0.75})$; where Y is the excretion of purine derivatives (mmol/day) and X is the microbial purines absorbed (mmol/day). The supply of microbial N in grams per day was calculated as follows: microbial N (g/day) = $(X × 70)/(0.116 × 0.83 × 1000) = 0.727 × X$; where X is the purine derivatives absorbed (mmol/day). The efficiency of microbial nitrogen synthesis (EMNS) was calculated using the equation of the Agricultural Research Council [20]: EMNS = [MN(g/day) × 1000 (g)]/DOMR (g); where DOMR = digestible organic matter apparently fermented in the rumen, DOMI = digestible organic matter intake, and DOMR = DOMI × 0.65. Nutrient digestibility was estimated using the acid insoluble ash technique according to Van Keulen and Young [21].

Milk yield was recorded daily, and milk samples were collected twice a day at 5:00 am and 03:00 pm for 7 days. Milk samples from each twice daily collection were mixed together at a 60:40 (v/v) ratio to make 100 ml of milk sample and stored at -20 C˚. After 7 days of collection, milk samples from each day and cow were mixed well, and a total of 100 ml was subsampled to analyze milk composition, including protein, fat, lactose, solids-not-fat, and total solids content, using Milko-Scan33 (Foss Electric, Hillerod, Denmark) at the Dairy Farming Promotion Organization of Thailand. The somatic cell count was analyzed using the Fossomatic 5000 Basic (Foss Electric, Hillerød, Denmark).

On the last day of the feeding trial, blood samples were collected from the jugular vein at 0 h before and 4 h post morning feeding. Eight milliliters of blood were collected from each cow and placed in tubes containing 12 mg of ethylenediaminetetraacetic acid. Blood samples were obtained by centrifugation at 500 × g for 10 min and used to analyze blood urea nitrogen according to Crocker [22]. The economic return for SBM *vs* CWYW top-dressing was calculated from the price of feed ingredients from the local market of Kranuan district, Khon Kaen province during the period of September 2020 to November 2020, where the price of rice straw was 0.07 US dollars per kg, concentrate feed cost 0.38 US dollars per kg, the SBM price was 0.61 US dollars per kg, and the CWYW price was 0.31 US dollars per kg. The price of milk was 0.59 US dollars per kg.

Ruminal fluid samples (100 mL) were carefully and quickly collected by a stomach tube connected to a vacuum pump on the last day of the feeding trial at 0 h before and 4 h post feeding. To avoid saliva contamination, the first three samples of ruminal fluid were discarded, and the pH of the ruminal fluid was immediately measured using a pH meter (HANNA Instruments HI 8424 microcomputer, Singapore). After measuring pH, fluid samples were separated into two parts: the first 45 mL of fluid samples were kept in a 60 ml plastic bottle containing 9 ml of 1 M $H_2SO_4$ and used to analyze for ammonia nitrogen ($NH_3$-N) concentration and volatile fatty acids (VFA), including acetate (C2), propionate (C3), and butyrate (C4) concentration. The second 1 ml of ruminal fluid samples were kept in 9 ml of 10% formalin and used to enumerate the bacterial and protozoal populations. Ruminal fluid samples were centrifuged at 16,000× g for 15 min and the supernatant was used for $NH_3$-N and VFA analysis. The concentration of $NH_3$-N was analyzed according to the standard method [15] using the distillation technique. Concentration of VFA was analyzed using gas chromatography (DB-Wax column- 30 m length, 0.25 mm diameter, 0.25 μm film; Agilent Technology, USA) as described by So et al. [23]. Bacterial and protozoal populations were counted on hemacytometers (Boeco, Hamburg, Germany) using microscopic according to Galyean [24].

## Statistical analysis

The following equation was used to evaluate the data as a completed randomized design (CRD) using SAS's (Version 9.4) Proc GLM procedure [25]:

$$Y_{ij} = \mu + \tau_i + \varepsilon_{ij}$$

where: $Y_{ij}$ = observations; $\mu$ = overall mean; $\tau_i$ = effect of treatment (SBM and CWYW and $\varepsilon_{ij}$ = the residual effect. Results are presented as mean values with the standard error of the mean. Differences between treatment means were determined by the Duncan's New Multiple Range Test [26] and differences of $p < 0.05$ were considered to represent statistically significant differences.

## Results

### Chemical compositions in the experimental diet

The basal diet contained CP, NDF, and ADF at 163, 198, and 143 g/kg DM, respectively (Table 1). When compared to SBM, CWYW contained more CP (520 *vs* 458 g/kg DM), NDF (392 *vs* 173 g/kg DM), and ADF (281 *vs* 133 g/kg DM). Moreover, CWYW costs less per kilogram than SBM (0.31 *vs* 0.61 US dollars).

### Feed intake, nutrient intake, and digestibility

Dietary intake, nutrient intake, and digestibility of cows fed with the SBM or CWYW topdress are shown in Table 2. Concentrate, supplements, rice straw, and total DM intake did not ($p > 0.05$) differ between SBM and CWYW top-dressing. CP intake was greater ($p < 0.05$) for the CWYW group. However, other nutrient intake and nutrient digestibility were not ($p > 0.05$) different between SBM and CWYW top-dressing.

### Ruminal pH, ammonia nitrogen, microbial count, and blood urea nitrogen

Ruminal ecology and BUN are reported in Table 3. Ruminal pH and the protozoal population did not ($p > 0.05$) differ between SBM and CWYW top-dressing. The mean ruminal pH was 6.6 in SBM and 6.7 in CWYW ($p > 0.05$). After 4 hours of feeding, CWYW top-dressing

**Table 2. Effect of replacing soybean meal with citric waste fermented yeast waste on feed intake and nutrient digestibility.**

| Item | SBM[1] | CWYW[2] | SEM | P-Value |
|---|---|---|---|---|
| Concentrates intake | | | | |
| kg/day | 8.5 | 8.4 | 0.29 | 0.83 |
| %BW daily | 1.7 | 1.7 | 0.06 | 0.88 |
| g/kg BW$^{0.75}$daily | 83.0 | 82.6 | 2.68 | 0.89 |
| Supplement intake | | | | |
| kg/day | 1.9 | 1.9 | 0.01 | 0.99 |
| %BW daily | 0.4 | 0.4 | 0.001 | 0.17 |
| g/kg BW$^{0.75}$daily | 18.7 | 18.7 | 0.08 | 0.99 |
| Rice straw intake | | | | |
| kg/day | 4.5 | 4.3 | 0.10 | 0.10 |
| %BW daily | 0.9 | 0.8 | 0.02 | 0.20 |
| g/kg BW$^{0.75}$daily | 43.9 | 41.9 | 0.95 | 0.11 |
| Total dry matter intake | | | | |
| kg/day | 14.9 | 14.6 | 0.36 | 0.55 |
| %BW daily | 3.2 | 3.1 | 0.06 | 0.52 |
| g/kg BW$^{0.75}$daily | 145.6 | 143.2 | 3.33 | 0.56 |
| Nutrient intake, kg/day | | | | |
| Organic matter | 13.1 | 12.9 | 0.33 | 0.57 |
| Crude protein | 2.3b | 2.5a | 0.03 | 0.03 |
| Neutral detergent fiber | 5.1 | 5.3 | 0.13 | 0.21 |
| Acid detergent fiber | 3.7 | 3.8 | 0.10 | 0.24 |
| Nutrient digestibility, g/kg | | | | |
| Dry matter | 687 | 686 | 0.20 | 0.26 |
| Organic matter | 715 | 713 | 1.08 | 0.18 |
| Crude protein | 760 | 761 | 0.51 | 0.75 |
| Neutral detergent fiber | 601 | 598 | 0.13 | 0.10 |
| Acid detergent fiber | 402 | 395 | 0.21 | 0.12 |

[a-b]Value on the same row with different superscripts differ ($p<0.05$).

*SEM*, standard error of mean;

[1]*SBM*, soybean meal;

[2]*CWYW*, citric waste fermented yeast waste.

showed greater ruminal $NH_3$-N, BUN, and bacterial population compared with the top-dress of SBM ($p<0.05$).

## Ruminal volatile fatty acid concentrations

Ruminal VFA concentrations are reported in Table 4. The total VFA, C2, C3, C4, and C2 to C3 ratios were not different ($p>0.05$) between cows fed the SBM or CWYW top-dressing. After 4 hours of feeding, total VFA was 108.3 mM in SBM and 107.7 mM in CWYW while C3 concentration was 29.5 mol/100 mol in SBM and 29.2 mol/100 mol in CWYW top-dressing.

## Urinary purine derivatives and microbial nitrogen supply

In Table 5, urinary purine derivatives and microbial protein synthesis are presented. Allantoin, creatinine, purine excretion, purine absorption, microbial nitrogen, MCP, and EMNS were

**Table 3. Effect of replacing soybean meal with citric waste fermented yeast waste on ruminal ecology and blood urea nitrogen.**

| Item | SBM[1] | CWYW[2] | SEM | P-Value |
|---|---|---|---|---|
| Ruminal pH | | | | |
| 0 h | 6.8 | 6.9 | 0.08 | 0.44 |
| 4 h | 6.4 | 6.4 | 0.07 | 0.75 |
| Mean | 6.6 | 6.7 | 0.05 | 0.41 |
| Ruminal $NH_3$-N, mg/dL | | | | |
| 0 h | 12.0 | 13.0 | 0.57 | 0.28 |
| 4 h | 22.0[b] | 25.6[a] | 0.63 | 0.01 |
| Mean | 17.0[b] | 19.3[a] | 0.37 | 0.01 |
| Blood urea nitrogen (BUN), mg/dL | | | | |
| 0 h | 9.0 | 10.0 | 0.57 | 0.28 |
| 4 h | 15.6[b] | 19.7[a] | 0.33 | 0.01 |
| Mean | 12.3[b] | 14.9[a] | 0.37 | 0.01 |
| Protozoal count, log10 cell/mL | | | | |
| 0 h | 5.91 | 5.89 | 0.044 | 0.79 |
| 4 h | 6.13 | 6.14 | 0.014 | 0.64 |
| Mean | 6.02 | 6.02 | 0.026 | 0.30 |
| Bacterial count, log10 cell/mL | | | | |
| 0 h | 9.91 | 9.90 | 0.032 | 0.82 |
| 4 h | 10.24[b] | 10.30[a] | 0.007 | 0.02 |
| Mean | 10.08 | 10.10 | 0.019 | 0.30 |

[a-b]Value on the same row with different superscripts differ (p<0.05).

*SEM*, standard error of mean;

[1]*SBM*, soybean meal;

[2]*CWYW*, citric waste fermented yeast waste.

not different (p>0.05) between cows fed SBM or CWYW top-dressing. The EMNS was 22.30 g nitrogen/kg OMDR for SBM and 22.85 g nitrogen/kg OMDR for CWYW, respectively.

## Milk production and compositions

Milk production, chemical composition, and economic return are presented in Table 6. The milk yield, 3.5% FCM, and milk composition were not different (p>0.05) between cows fed the SBM or CWYW top-dressing. MUN was greater (15.8 *vs* 14.7 mg/dL) and SCC was lower (4.94 *vs* 5.09 log10 cell/mL) in milk from cows fed the CWYW top-dress compared to milk from cows fed the SBM top-dress (p<0.05). Top-dressed cost (0.59 *vs* 1.16 US dollars/cow/day) and total feed cost (4.10 *vs* 4.71 US dollars/cow/day) were lower (p<0.05) for CWYW compared to SBM. The milk sale and margin over feed cost were not different (p>0.05) between SBM and CWYW top-dressing.

## Discussion

Top-dressed CWYW has no impact on the total DM intake compared with top-dressed SBM. Similarly, Uriyapongson et al. [27] supplemented 10% of citric waste in a concentrate and feed intake was not affected. Cherdthong et al. [9] indicated that yeast waste could substitute 100% of SBM without adverse effects on DM intake in beef cattle. The greater CP intake for cows fed CWYW compared with SBM top-dressing may be due to the greater CP content in CWYW than in SBM [14, 28].

**Table 4. Effect of replacing soybean meal with citric waste fermented yeast waste on volatile fatty acids (VFAs).**

| Item | SBM[1] | CWYW[2] | SEM | P-Value |
|---|---|---|---|---|
| Total VFA, *mM* | | | | |
| 0 h | 91.5 | 90.9 | 2.74 | 0.92 |
| 4 h | 108.3 | 107.7 | 1.84 | 0.89 |
| Mean | 99.9 | 99.3 | 1.90 | 0.89 |
| Acetic acid, mol/100 mol | | | | |
| 0 h | 65.1 | 65.2 | 0.52 | 0.93 |
| 4 h | 61.3 | 61.5 | 0.59 | 0.79 |
| Mean | 63.2 | 63.4 | 0.35 | 0.75 |
| Propionic acid, mol/100 mol | | | | |
| 0 h | 25.7 | 25.4 | 0.32 | 0.57 |
| 4 h | 29.5 | 29.2 | 0.43 | 0.64 |
| Mean | 27.6 | 27.3 | 0.17 | 0.27 |
| Butyric acid, mol/100 mol | | | | |
| 0 h | 9.2 | 9.4 | 0.39 | 0.69 |
| 4 h | 9.2 | 9.3 | 0.35 | 0.84 |
| Mean | 9.2 | 9.3 | 0.34 | 0.79 |
| Acetic acid to Propionic acid Ratio | | | | |
| 0 h | 2.5 | 2.6 | 0.05 | 0.37 |
| 4 h | 2.1 | 2.1 | 0.01 | 0.99 |
| Mean | 2.3 | 2.3 | 0.02 | 0.37 |

[a-b]Value on the same row with different superscripts differ (p<0.05). *SEM*, standard error of mean;

[1]*SBM*, soybean meal;

[2]*CWYW*, citric waste fermented yeast waste.

In our *in vitro* study, Suriyapha et al. [14] showed that SBM replaced with CWYW at 100% decreased *in vitro* dry matter digestibility (0.80%). However, in this study, nutrient digestibility did not differ between SBM and CWYW top-dressing. This could be due to the fact that, the experimental animals had no significant difference in DM, NDF, and ADF intake. As a result,

**Table 5. Effect of replacing soybean meal with citric waste fermented yeast waste on urinary purine derivative and microbial protein synthesis.**

| Item | SBM[1] | CWYW[2] | SEM | P-Value |
|---|---|---|---|---|
| Purine derivative, mmol/d | | | | |
| Allantoin | 159.88 | 160.66 | 0.31 | 0.27 |
| Excretion | 192.84 | 193.78 | 0.95 | 0.24 |
| Absorption | 186.77 | 187.88 | 0.73 | 0.21 |
| Creatinine, mg/dL | 59.72 | 59.62 | 0.08 | 0.71 |
| Microbial N, g/d | 135.78 | 136.59 | 0.97 | 0.21 |
| MCP[3], g/d | 848.44 | 853.37 | 1.83 | 0.09 |
| EMNS[4], gN/kg OMDR | 22.30 | 22.85 | 0.21 | 0.07 |

[a-b]Value on the same row with different superscripts differ (p<0.05). *SEM*, standard error of mean;

[1]*SBM*, soybean meal;

[2]*CWYW*, citric waste fermented yeast waste;

[3]*MCP*, microbial crude protein;

[4]*EMNS*, efficiency of microbial N synthesis = [MN(g/day) × 1000 (g)]/DOMR (g);

where DOMR = DOMI × 0.65, DOMR = digestible organic matter apparently fermented in the rumen and DOMI = digestible organic matter intake [20].

**Table 6. Effect of replacing soybean meal with citric waste fermented yeast waste on milk production, chemical composition and economic return.**

| Item | SBM[1] | CWYW[2] | SEM | P-Value |
|---|---|---|---|---|
| Milk Yield, kg/day | 16.8 | 16.4 | 0.94 | 0.76 |
| 3.5% FCM[3], kg/day | 16.8 | 16.4 | 0.87 | 0.71 |
| Milk composition, g/kg | | | | |
| Fat | 35.2 | 35.0 | 0.12 | 0.94 |
| Protein | 34.2 | 34.3 | 0.08 | 0.89 |
| Lactose | 45.3 | 44.7 | 0.16 | 0.75 |
| Solids-not-fat | 87.7 | 87.0 | 0.25 | 0.61 |
| Total solids | 122.9 | 122.0 | 0.36 | 0.86 |
| Milk urea nitrogen (MUN), mg/dL | 14.7[b] | 15.8[a] | 0.32 | 0.04 |
| Somatic cell count (SCC), log10 cell/mL | 5.09[a] | 4.94[b] | 0.04 | 0.04 |
| Economic return (US dollar [4]/cow/day) | | | | |
| Feed cost | | | | |
| Roughage cost | 0.32 | 0.31 | 0.01 | 0.12 |
| Concentrate cost | 3.23 | 3.20 | 0.11 | 0.90 |
| Top-dressed cost | 1.16[a] | 0.59[b] | 0.01 | <0.01 |
| Total feed cost | 4.71[a] | 4.10[b] | 0.12 | <0.01 |
| Milk sale | 9.91 | 9.68 | 0.56 | 0.97 |
| Margin over feed cost | 5.20 | 5.58 | 0.49 | 0.39 |

[a-b]Value on the same row with different superscripts differ (p<0.05).

*SEM*, standard error of mean;

[1]*SBM*, soybean meal;

[2]*CWYW*, citric waste fermented yeast waste;

[3]*3.5% FCM*, fat collected milk = 0.432×(kg of milk/d)+16.23×(kg of fat) [53];

[4] *US dollar* = 31.16 Thai baht [78].

there was no difference in digestibility. In particular, the NDF and ADF have a major impact on feed intake restrictions and are responsible for dietary digestibility limitations [4, 29, 30]. Furthermore, it could be related to appropriate media solutions and yeast cells from yeast waste in CWYW, which are rich in amino acids, minerals, and vitamins [9, 14]. It improved feed nutritional values and helped to promote rumen microorganisms, especially the rumen cellulolytic, amylolytic, and proteolytic bacteria, which improved the digestibility of nutrients and rate of digestion [31, 32]. Similarly, Cherdthong et al. [9] found that when yeast waste replaced SBM at 100% as a protein source in the concentrate, there was no effect on nutrient digestibility in beef cattle.

The pH is an important factor that determines the activity of ruminal microbes [33]. In this study, the mean pH was higher in CWYW than in SBM top-dressing; 6.7 *vs* 6.6. This is probably due to yeast from CWYW not only encouraging lactate users and increasing their number, but also competing with lactate producers [34]. In our *in vitro* study, Suriyapha et al. [14] found that the replacement of SBM with CWYW had no negative effect on ruminal pH. Similarly, Cherdthong et al. [9] revealed that when yeast waste replaced 100% of soybean meal, there was no negative impact on the ruminal pH of Thai native cattle.

The greater ruminal $NH_3$-N concentration at 4 h after feeding and the greater mean value for CWYW top-dressing compared to SBM top-dressing could be due to the greater CP content and CP digestibility of CWYW compared to SBM. Greater CP digestibility may have occurred as a result of the microbial disintegration of yeast cells [35] because of changes in the

number of ruminal microorganisms with proteolytic activity [36] or to the ability to supply substances as stimulants, including protein, to bacteria in the rumen [37, 38]. Another effect is the high level of NPN-urea in the CWYW, which causes increased levels of urea-N in the diet and the fast hydrolysis of NPN-urea into ruminal $NH_3$-N by microbial enzymes, causing an elevation in ruminal $NH_3$-N concentration [28, 39]. Similarly, Suriyapha et al. [14] found that the *in vitro* ruminal $NH_3$-N concentration was increased when higher levels of CWYW replaced soybean meal. Normally, a minimum of $NH_3$-N at 5 mg/dL is sufficient for proper microbial synthesis in the rumen [40]. Because both treatments in the current study were well above this concentration, we may conclude that the rumen microbiomes had enough $NH_3$-N to support microbial protein synthesis.

Normally, blood urea nitrogen (BUN) values in tropical ruminants range from 6.3 to 25.5 mg/dl, depending on diet and feeding pattern [30]. In this study, the BUN at 4 h post-feeding and the mean value were higher in CWYW than in SBM top-dressing. This could be due to the greater ruminal $NH_3$-N concentration in CWYW than in SBM top-dressing. Patra and Aschenbach [41] revealed that the BUN concentration is related to the level of ruminal $NH_3$-N production. Urea is 100% ruminal degradable because it can be swiftly converted into ammonia by rumen ureolytic bacteria [42, 43]. Furthermore, Xu et al. [44] demonstrated that the amount of $NH_3$-N absorbed from the rumen is reflected in circulating BUN with incremental urea supplementation.

Total ruminal bacteria at 4 h post-feeding was higher in CWYW than in SBM top-dressing. *S. cerevisiae* in CWYW is the essential source of peptides, amino acids, β-glucan, sugar, and vitamins, which could stimulate the rumen bacterial population [38, 45, 46]. Similarly, Díaz et al. [47] showed that adding yeast hydrolysate promoted ruminal microbe growth when compared to the non-supplemented group. Chaucheyras-Durand et al. [34] reported that yeast supplementation induced significant changes in relative abundances of a few bacterial species, especially *Fibrobacter succinogenes* in lambs. The number of protozoa was not different between SBM and CWYW top-dressing. Our *in vitro* study [14] also found that substitution of soybean meal with CWYW did not affect *in vitro* protozoa populations. Similarly, Cherdthong et al. [9] reported that the substitution of soybean meal with yeast waste showed no effect on the number of protozoa in cattle. Galip [48] and Chaucheyras-Durand et al. [49] found that *S. cerevisiae* did not affect the protozoa count, which could be due *to S. cerevisiae* competing with protozoa for sugar utilization [49].

Yeast cells have been shown to improve volatile fatty acid (VFA) profiles through modifying mainly the production of C2 and C3 [47, 50, 51]. In this study, total VFA and molar portions of VFA were not different between SBM and CWYW top-dressing. This is probably due to the non-significance in dry matter and organic matter digestibility between SBM and CWYW top-dressing. Increased digestibility indicates increased utilization of nutrients by microorganisms, which increases the microbial fermentation end-products [52]. Suriyapha et al. [14] found that SBM could be replaced by CWYW in concentrate diets without a negative impact on VFA profiles. Similarly, Cherdthong et al. [9] reported that the substitution of SBM with yeast waste showed no effect on VFA and molar portions of VFA.

Purine derivative (PD) excretion in the urine is a biomarker used to estimate rumen microbial activity and productivity. Ruminal microbial protein synthesis is essential in ruminants as it supplies great-quality and essential protein sources for the ruminant animals [42, 53]. The great potential of microbial protein production from rumen microbes can be enhanced by the provision of a high-quality feed source [54]. However, several factors influence microbial protein synthesis efficiency, including DM intake, ruminal nitrogen and carbohydrate degradability rates, mineral and vitamin content, and other factors [53]. In the present study, allantoin, purine derivative excretion, and EMNS were not different between SBM and CWYW top-

dressing. This is most likely due to yeast waste in CWYW supplying essential growth or the ability to supply stimulatory substances to rumen microbes [38, 45]. When *S. cerevisiae* is anaerobically fermented, it can synthesize more or produce amino acids and organic acids [55, 56]. Microbial growth was most likely aided by higher ruminal $NH_3$-N levels. Ammonia is the major N source of the ruminal microorganism and the diet accessibility of peptides and amino acids for enhancing cellulolytic and amylolytic bacteria's growth. [13, 53]. Russell and Rychlik [57] revealed that ammonia nitrogen released from protein degradation is used in ruminal microbial protein synthesis, which would advocate the outflow of protein into the lower gut, which would supply the animal with more protein [42, 53]. Moreover, the higher CP intake with supplemented CWYW could have enhanced the protein flow from the rumen to the small intestine [58]. Similarly, Zhu et al. [59] revealed that the number of microbial proteins produced was enhanced in yeast cell-supplemented cows.

Milk production and composition were not different between SBM and CWYW top-dressing. This is probably due to the fact that CWYW has the potential to replace the use of soybean meal in animal feeds [14]. It is known that lactating cows have a high nutritional demand for milk and milk protein production, which causes body protein mobilization to provide amino acids for milk protein synthesis and also gluconeogenesis [60, 61]. Thus, feeding more protein is speculated to boost milk production by supplying more amino acids for gluconeogenesis, or the amino acids required for lipoprotein synthesis. Ghelichkhan et al. [7] reported that a top-dress of 2 kg of solvent SBM could increase milk output with no negative impact on milk composition. In our study, CWYW contains yeast cells from yeast waste obtained from ethanol processes with 360 g/kg of crude protein and is rich in amino acids, minerals, and vitamins [14, 39, 46, 47]. Additionally, yeast cells can increase milk production by 3 to 9% in cows supplemented with yeast [59, 62]. Yeast cells are rich in nutrients, especially true protein and amino acids, that benefit the milk production process and the ruminal fermentation process as well [56]. In addition, another factor that may be CWYW contains urea, where urea is converted to ammonia, the main N source for rumen microorganisms. The microbial protein synthesis using ammonia-N released from the rumen would potentially increase the outflow of protein into the lower gut, which would supply the animal with more protein and essential amino acids [53].

Milk urea nitrogen (MUN) is a by-product of protein metabolism as an indicator of protein nutrient status and N utilization efficiency in dairy cows [63]. In this study, top-dressed CWYW showed greater MUN than SBM top-dressing. This could be due to the greater BUN in CWYW than in SBM top-dressing. Thus, BUN could partition into the milk [63, 64]. The concentration of MUN has a direct impact on BUN concentration as a result of urea diffusing readily from the bloodstream into milk [65]. The MUN concentrations in this study were close to the optimal MUN range of tropical dairy cows fed with urea-treated rice straw (11.4 to 17.5 mg/dL) [3, 64, 66]. In contrast, previous studies have demonstrated that dietary yeast supplementation has no effect on MUN in lactation cows [67, 68].

Somatic cell count (SCC) is the common parameter for demonstrating defense mechanisms against sickness and infection of the mammary gland in dairy cows [68]. In this study, we found that the SCC in milk of the CWYW group was decreased, which might be due to yeast cell walls from the yeast waste improving the local systemic immune responses and inhibiting infection [56, 68, 69]. The stimulatory effect of yeast may be due to the antigen-presenting ability of β-1,3-glucan in yeast cell walls [67]. β-Glucan has been identified as a pathogen-associated molecular pattern (PAMP) that primes T cells, resulting in enhanced proliferation and response to antigens or cytokines [70]. Additionally, mannan oligosaccharide (MOS) can operate as an affinity ligand that binds with mannose-specific type-1 fimbriae on gram-negative bacteria [71, 72]. This mechanism may elicit substantial antigenic responses, boosting

humoral immunity against specific pathogens. An increased humoral immunity response and better defense against infection has the potential to decrease the pro-inflammatory immune response, which is negative for production [68, 73–75]. Similarly, Stefenoni et al. [76] found a reduction in subclinical mastitis and several clinical cases of mastitis in cows supplemented with hydrolyzed yeast. However, the use of β-glucans in dirty quarters had no effect on chronic subclinical mastitis or reduced SCC in lactating cows, which could be a lack of local immunity stimulation and systemic [77].

## Conclusions

Top-dressed CWYW at 0.4% of BW was comparable with SBM top-dressing in terms of intake, VFA, purine derivatives, milk yield, and milk composition. CWYW top-dressing showed greater CP digestibility, $NH_3$-N, BUN, and MUN while having lower SCC when compared with SBM top-dressing. Hence, CWYW could be used as a protein source ingredient and reduce the production costs of farmers while also providing environmental benefits as a zero-waste system. However, further long-term studies (entire 305-day lactation) are necessary to determine production responses in tropical lactating dairy cows.

## Acknowledgments

The authors would like to express our sincere thanks to the Tropical Feed Resources Research and Development Center (TROFREC), Department of Animal Science, Faculty of Agriculture, Khon Kaen University (KKU) for the use of the research facilities and making this work possible. The authors acknowledge the KSL Green Innovation Public Co., Ltd and Sukkhee Farm, Thailand. Moreover, the authors are sincerely grateful to acknowledge J. P. Schoonmaker (Associate Professor in the Department of Animal Sciences, Purdue University, USA) for his help in recommending, suggesting, and proofreading our manuscript.

## Author Contributions

**Conceptualization:** Chaichana Suriyapha, Anusorn Cherdthong.

**Data curation:** Chaichana Suriyapha, Chanadol Supapong, Sarong So.

**Formal analysis:** Chaichana Suriyapha.

**Funding acquisition:** Chaichana Suriyapha, Anusorn Cherdthong.

**Investigation:** Chaichana Suriyapha, Chanadol Supapong, Sarong So.

**Methodology:** Chaichana Suriyapha, Chanadol Supapong, Sarong So.

**Project administration:** Chaichana Suriyapha, Anusorn Cherdthong.

**Resources:** Chaichana Suriyapha, Anusorn Cherdthong.

**Software:** Chaichana Suriyapha, Chanadol Supapong, Sarong So.

**Supervision:** Metha Wanapat, Anusorn Cherdthong.

**Validation:** Chaichana Suriyapha, Chanadol Supapong, Sarong So, Metha Wanapat, Anusorn Cherdthong.

**Writing – original draft:** Chaichana Suriyapha.

**Writing – review & editing:** Chaichana Suriyapha, Chanadol Supapong, Sarong So, Metha Wanapat, Anusorn Cherdthong.

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
