## [Decision Letter · Decision Letter 0]

18 Jul 2022

PONE-D-22-12915Bioconversion of agro-industrial residues as a protein source supplementation for multiparous Holstein Thai crossbreed cowsPLOS ONE

Dear Dr. Cherdthong,

Thank you for submitting your manuscript to PLOS ONE. After careful consideration, we feel that it has merit but does not fully meet PLOS ONE’s publication criteria as it currently stands. Therefore, we invite you to submit a revised version of the manuscript that addresses the points raised during the review process.

Please submit your revised manuscript by Sep 01 2022 11:59PM. If you will need more time than this to complete your revisions, please reply to this message or contact the journal office at plosone@plos.org. Please include the following items when submitting your revised manuscript:A rebuttal letter that responds to each point raised by the academic editor and reviewer(s). You should upload this letter as a separate file labeled 'Response to Reviewers'.A marked-up copy of your manuscript that highlights changes made to the original version. You should upload this as a separate file labeled 'Revised Manuscript with Track Changes'.An unmarked version of your revised paper without tracked changes. You should upload this as a separate file labeled 'Manuscript'.If applicable, we recommend that you deposit your laboratory protocols in protocols.io to enhance the reproducibility of your results. Protocols.io assigns your protocol its own identifier (DOI) so that it can be cited independently in the future. For instructions see: https://journals.plos.org/plosone/s/submission-guidelines#loc-laboratory-protocols. Additionally, PLOS ONE offers an option for publishing peer-reviewed Lab Protocol articles, which describe protocols hosted on protocols.io. Read more information on sharing protocols at https://plos.org/protocols?utm_medium=editorial-email&utm_source=authorletters&utm_campaign=protocols.

We look forward to receiving your revised manuscript.

Kind regards,

Mahmoud A.O. Dawood, PhD

Academic Editor

PLOS ONE

Journal Requirements:

“-CS, AC

-RRI-PHD62I0021

-Research and Researchers for Industries

-http://rri.trf.or.th/bs_detail.asp

-Funders did not play any role in the study design, data collection and analysis, decision to publish, or preparation of the manuscript.”

 “The authors would like to express our sincere thanks to the Animal Feed Inter Trade 398 Co., Ltd and Thailand Science Research and Innovation (TSRI) through the Research and 399 Researcher for Industry (RRi) program (contract grant PHD62I0021) for the scholar funding 400 and research grant”

Reviewers' comments:

Reviewer's Responses to Questions

**Comments to the Author**

1. Is the manuscript technically sound, and do the data support the conclusions?

Reviewer #1: Yes

2. Has the statistical analysis been performed appropriately and rigorously? 

Reviewer #1: Yes

3. Have the authors made all data underlying the findings in their manuscript fully available?

Reviewer #1: Yes

4. Is the manuscript presented in an intelligible fashion and written in standard English?

Reviewer #1: Yes

5. Review Comments to the Author

Reviewer #1: Overall, this study revealed the huge potential of industry waste to be use as alternative protein source for dairy cattle.

However, this paper need to provide further information as follow:

Introduction

1. Need to provide more information to justify why using CWYW to alternate SBM - any economical value?

Materials and Methods

1. Please mention yeast species

6. PLOS authors have the option to publish the peer review history of their article (what does this mean?). If published, this will include your full peer review and any attached files.

Reviewer #1: No

---

## [Author Response · Author response to Decision Letter 0]

19 Jul 2022

Response: Thank you, we have done it. Please see below.

Response: Thank you. We have modified according to the style requirement. Please see in manuscript.

“-CS, AC

-RRI-PHD62I0021

-Research and Researchers for Industries

-http://rri.trf.or.th/bs_detail.asp

-Funders did not play any role in the study design, data collection and analysis, decision to publish, or preparation of the manuscript.”

 “The authors would like to express our sincere thanks to the Animal Feed Inter Trade 398 Co., Ltd and Thailand Science Research and Innovation (TSRI) through the Research and 399 Researcher for Industry (RRi) program (contract grant PHD62I0021) for the scholar funding 400 and research grant”

Response: Thank you. The acknowledgments have been modified as: 

The authors would like to express our sincere thanks to the Thailand and Tropical Feed Resources Research and Development Center (TROFREC), Department of Animal Science, Faculty of Agriculture, Khon Kaen University (KKU) for the use of the research facilities and making this work possible. The authors acknowledge the KSL Green Innovation Public Co., Ltd, Sukkhee Farm, Thailand. Moreover, the authors are sincerely grateful to acknowledge J. P. Schoonmaker (Associate Professor in the Department of Animal Sciences, Purdue University, USA) for his help in recommending, suggesting, and proofreading our manuscript.

 In addition, the funding has been removed from the manuscript and please publish funding information as

Funding

This work was supported by Animal Feed Inter Trade Co., Ltd, Thailand Science Research and Innovation (https://www.tsri.or.th) through the Research and Researcher for Industry (RRi) (http://rri.trf.or.th/bs_detail.asp) program (contract grant PHD62I0021), Research Program on the Research and Development of Winged Bean Root Utilization as Ruminant Feed, Increase Production Efficiency and Meat Quality of Native Beef and Buffalo Research Group and Research and Graduate Studies, Khon Kaen University (KKU), and Office of National Higher Education Science Research and Innovation Policy Council (https://www.nxpo.or.th/) through the Program Management Unit for Competitiveness (PMUC) (https://www.nxpo.or.th/) (contract grant C10F640078). Funders did not play any role in the study design, data collection and analysis, decision to publish, or preparation of the manuscript.

Response: We have checked and modified.

Response to Reviewer 1

Introduction

1. Need to provide more information to justify why using CWYW to alternate SBM - any economical value? 

Response: Thank you for your suggestion. We have provided as “When comparing economic values, CWYW was cheaper than SBM by 50% per kilogram (0.31 USD vs. 0.61 USD).” Please see in line 66-67. In addition, we have provided detail of feed cost and CWYW cost in the results section and Table 1. Please see in manuscript.

Materials and Methods

1. Please mention yeast species

Response: Thanks for your suggestion. Present study we used Saccharomyces cerevisiae. We have added as “A by-product of ethanol manufacturing called yeast waste, which includes inoculants of the yeast (Saccharomyces cerevisiae), was obtained from Green Innovation Public Company Limited (KGI) a subsidiary of Khon Kaen Sugarcane Industry Company Limited in Nam Phong district, Khon Kaen province, Thailand.” Please see in line 82-85.

Thanks so much for the valuable comments!

---

## [Editor Report · Decision Letter 1]

18 Aug 2022

Bioconversion of agro-industrial residues as a protein source supplementation for multiparous Holstein Thai crossbreed cows

PONE-D-22-12915R1

Dear Dr. Cherdthong,

We’re pleased to inform you that your manuscript has been judged scientifically suitable for publication and will be formally accepted for publication once it meets all outstanding technical requirements.

Kind regards,

Mahmoud A.O. Dawood, PhD

Academic Editor

PLOS ONE
---

## [Editor Report · Acceptance letter]

22 Aug 2022

PONE-D-22-12915R1 

Bioconversion of agro-industrial residues as a protein source supplementation for multiparous Holstein Thai crossbreed cows 

Dear Dr. Cherdthong:

I'm pleased to inform you that your manuscript has been deemed suitable for publication in PLOS ONE. Congratulations! Your manuscript is now with our production department. 

Kind regards, 

on behalf of

Dr. Mahmoud A.O. Dawood 

Academic Editor

PLOS ONE